# The Novel Combination of Nano Vector Network Analyzer and Machine Learning for Fruit Identification and Ripeness Grading

**DOI:** 10.3390/s23020952

**Published:** 2023-01-13

**Authors:** Van Lic Tran, Thi Ngoc Canh Doan, Fabien Ferrero, Trinh Le Huy, Nhan Le-Thanh

**Affiliations:** 1Universite Cote d’Azur, LEAT, CNRS, 06903 Sophia Antipolis, France; 2The University of Danang—University of Science and Technology, Danang 550000, Vietnam; 3The University of Danang—University of Economics, Danang 55000, Vietnam; 4The Univeristy of Danang, Danang International Institute of Technology—DNIIT, Danang 550000, Vietnam; 5Faculty of Computer Engineering, University of Information Technology, Ho Chi Minh City 721400, Vietnam

**Keywords:** VNA, KNN, neural network, fruit classification

## Abstract

Fruit classification is required in many smart-farming and industrial applications. In the supermarket, a fruit classification system may be used to help cashiers and customer to identify the fruit species, origin, ripeness, and prices. Some methods, such as image processing and NIRS (near-infrared spectroscopy) are already used to classify fruit. In this paper, we propose a fast and cost-effective method based on a low-cost Vector Network Analyzer (VNA) device augmented by K-nearest neighbor (KNN) and Neural Network model. S-parameters features are selected, which take into account the information on signal amplitude or phase in the frequency domain, including reflection coefficient *S*_11_ and transmission coefficient *S*_21_. This approach was experimentally tested for two separate datasets of five types of fruits, including Apple, Avocado, Dragon Fruit, Guava, and Mango, for fruit recognition as well as their level of ripeness. The classification accuracy of the Neural Network model was higher than KNN with 98.75% and 99.75% on the first dataset, whereas the KNN was seen to be more effective in classifying ripeness with 98.4% as compared to 96.6% for neural network.

## 1. Introduction

Classification and the grading of fruits and vegetables play an important role in the post-harvest procedure. Traditionally, farmers and distributors perform this step through human operators or human senses such as handpicking or touching. Such a tedious, time-consuming, slow, and inconsistent method results in huge post-harvest losses. Another application of post-harvest handling is identifying immature and over-mature fruits, which is crucial to their subsequent storage, marketable life, and high-quality products to customers. Indeed, in some fruits it is difficult to recognize ripeness according to appearance before cutting the fruit, such as watermelon [1]. For these reasons, the increased demand for an accurate, fast, and nondestructive technique of agricultural products has increased in recent years. Many approaches have been introduced in the literature for the automated inspection of fruit types, ripening stage identification [1] or detection, and the classification of citrus diseases [2,3].

The ratio of the intensity of peak1 to peak2 (RPP) and the normalized difference intensity of peak (NDIP) methods [4] employ the visible and near-infrared spectroscopy (Vis/NIR) technology to determine the maturity level of watermelon. The Correct-RPP (C-RPP) method presents little better accuracy than NDIP with CCR of 85.1%, and much better than the LS-SVM classifier with CCR of 76.7% [5]. Ref. [6] proposed two families of deep neural networks EfficientNet and MixNet for automated fruit recognition. This system enhanced the prediction outcome compared to a well-established baseline and assured the real-time recommendations [6]. Among these classifiers, the best accuracy of 99.98% and 99.97% were respectively obtained by MixNetSmall and MixNetSmall, among its counterparts. A dataset of 48,905 images for training and 16,421 images for testing was used to evaluate the performance of this approach. A fuzzy model based on the Classification and Regression Tree (CART) algorithm was introduced by Ref. [7] for grading the ripening stages of bananas. Two simple features, namely peak hue and normalized brown area (NBA), were employed as inputs for this model. An accuracy of 93.11% was achieved by evaluating the proposed method on the MUSA database including banana samples at different ripening stages. The authors in Ref. [8] investigated two deep learning architectures, a small CNN model, and a VGG-16 fine-tuned model for fruit classification. The approach’s performance has been validated on two datasets: dataset 1 of supermarket produce [9] and dataset 2 of images collected from the Internet. Classification accuracies of 99.49% and 99.75% were achieved by evaluating the proposed models on dataset 1, while the reported accuracy for these two models was 85.43% and 96.75%, respectively.

The novel approach in Ref. [10] relied on an ImageNet pre-trained convolutional neural network to classify different types of fruits and vegetables. This fine-tuned classifier includes relevant spatial and spectral features extracted from the hyperspectral images. The additional information provided by this dataset allows to improve the overall accuracy of 85.22% and the average accuracy of 88.15%. However, the previous classification techniques have one or more shortcomings, such as:The image analysis may not be robust because different fruit images may have indistinguishable shape and color features. Some fruits (e.g., some varieties of peaches, apples, and watermelons) do not change their skin color as the fruit ripens or matures. In this case, the global color cannot be used as a ripeness evaluation parameter;Hyperspectral imaging-based techniques take a great deal of computational time because of their complexity. A hyperspectral image includes a large number of bands whose information is frequently highly correlated, leading to a huge amount of data;Near-infrared spectroscopy-based methods depend highly on position measurement, the surface of objects, as well as environmental white light.

Recently, several non-invasive methods based on radio-frequency have been investigated for fruit/vegetable classification to solve previous limitations. A contact-less measurement of permittivity was proposed in Ref. [11] for agricultural applications. This method can be a first step for the quality assessment of fruit but requires further development to demonstrate it. In Ref. [12], quality assessment was performed at terahertz on thin fruit slices. Even if the technique is non-invasive, the fruit has to be cut into slices, which limits its application. In Ref. [13], a circular antenna array with 10 elements was used to determine the maturity of watermelon. The imaging method is accurate but it requires a large and complicated feeding mechanism.

To overcome these shortcomings, this paper introduces an efficient method based on the combination between a low-cost Vector Network Analyzer (VNA) and machine learning algorithms for two principal applications: fruit classification and fruit maturity prediction. However, there has been no research on the combined use of low-cost VNA and basic machine learning models, such as the neural network model or K-nearest neighbor, for the identity recognition and freshness detection of fruits. Therefore, this study aims to explore the potential that these two technologies can be used to determine fruit type as the first step of the fruit freshness detection problem. This paper is organized as follows. Section 1 reviews related works on low-cost VNA and machine learning applications in food and agriculture and the motivation of this study. The collection of VNA data is presented in Section 2.1. Section 2.2 and Section 2.3 describe our proposed solutions, including feature extraction methods. Section 2.4 describes pre-processing and two machine learning algorithms. The experimental results are given in Section 3. Section 4 presents the conclusion and future work of the current research.

## 2. Proposed Method

Our predictive models are designed by using supervised machine learning algorithms, including two steps. In the training step, the spectra of 40 data points that were swept from 5 types of fruits were used directly. After that, the S-parameters are selected as relevant features for training the predictive models. The testing phase employed test data and the same feature extraction as the training phase. The details of these technical steps are described in Figure 1 and discussed below.

### 2.1. Data Collection

The main device used in data collection is NanoVNA, which is a tiny handheld Vector Network Analyzer (VNA), designed by edy555. It is a very portable but high-performance vector network analyzer. It is a standalone portable device with an LCD and battery. NanoVNA was designed to work at 50kHz-1500MHz. Calibration is also needed before measuring with NanoVNA.

The collected data are S-parameters, which can describe the input-output relationship between ports (or terminals) in an electrical system. In NanoVNA, we have 2 ports (called Port 0 and Port 1). In Port 0, the most commonly quoted parameter concerning antennas is *S*_11_. *S*_11_ represents how much power is reflected from the antenna and hence is known as the reflection coefficient (sometimes written as gamma, or return loss). The remainder of the power is “accepted by” or delivered to the antenna. The accepted power is either radiated or absorbed as losses within the antenna. Since antennas are typically designed to be low loss, ideally the majority of the power delivered to the antenna is radiated. In Port 1, the *S*_21_ represents the power transferred from Port 0 to Port 1, which is known as the transmission coefficient. The measurement device and antenna are shown in Figure 2.

To get $S_{11}$ and $S_{21}$, we use nanoVNASaver software, which was developed by Rune B. Broberg. It is a multi-platform tool used to save Touchstone files from the NanoVNA. This software connects to a NanoVNA, then extracts the data and saves Touchstone files into the computer.

Two antennas are required to perform the reflection and transmission measurements. The main specification for handheld near-field sensing antenna are sensitivity, polarization and compactness. Circular polarization is desired to avoid polarization alignment mismatch. Inverted F Antenna (IFA) are known to provide a compact form factor with linear polarization. Circular polarization can be obtained with a triple and quadruple layout with the rotational axis of symmetry and a suitable feeding network. A tri-fillar antenna has been designed to radiate Right Hand Circular Polarization (RHCP) in the UHF band with an ultra-small form factor by RFthings company for communication with a satellite [14]. The antenna provides a 110° wide beam angle RHCP radiation. A 3 dBic peak gain is obtained in the broadside direction. The antenna is connected through an SMA Male or a 100 mm flexible cable using a U.FL connector. The structure is available for three different frequency bands: 868, 915, or 923 MHz. In this measurement, we use a frequency band at 923 MHz, which corresponds to the ISM band in Vietnam. Figure 3 shows the data collection model.

The white box made of wood, as shown in Figure 4, was designed as a proper place not only to put fruit inside but also to adjust the distance between the two antennas and fruits corresponding to the size of each fruit. In this measurement, NanoVNA is set with a sweep from 700 MHz to 1 GHz with sweep frequency spans in segments to retrieve 40 data points.

### 2.2. Feature Extraction

As presented in the previous section, $S_{11}$ and $S_{12}$ are scattering parameters that measure how radio frequency energy transmits or reflects through a multi-port system. Thus, we propose to construct the S-parameters feature vector to present the reflection/transmission capacity quantified by amplitude and optionally phase in the frequency domain. Such features can be used in various shape analysis problems such as fruit ripeness detection [15]. Figure 5 illustrates the difference between $S_{11}$ and $S_{12}$ for two types of fruits with different inside and outside characteristics, such as guava and tomato.

In this measurement, we use the *.s2p format, which includes Stim(Frequency), Real(*S*_11_), Imag(*S*_11_), Real(*S*_21_), and Imag(*S*_21_). Although S-Parameters ($S_{11}$ and $S_{21}$) are provided with Real and Imaginary complex values, they can be easily converted to Magnitude and Phase information as the feature. Consequently, the full feature vector is the combination of four parts including Real, Image, Magnitude, and Phase of $S_{11}$ and $S_{21}$.

The formulas for calculating the Magnitude and Phase from the Real and Image numbers are:(1)$$S\left(Magnitude\right)\left[dB\right]=20\log\left(\sqrt{S(Real)+S(Image)}\right)$$
(2)$$S\left(Phase\right)=\arctan\left(\frac{S(Image)}{S(Real)}\right)$$

Figure 6 describes the Magnitude, Phase, Real, and Image of *S*_11_ and *S*_21_ of guava.

Figure 7 shows the magnitude of *S*_11_ and *S*_21_ for five kinds of fruits: Guava, Tomato, Orange, Avocado, and Mango. In this figure, 10 samples of each fruit are plotted in order to illustrate measurement variability. The fruits are clearly distinguished by the line group colors with *S*_11_ and *S*_21_ Magnitude; it is clearly visible in the zoom area. They also converge fruits into line groups. However, the result also has a line that is difficult to distinguish, such as *S*_21_ of Avocado (black line).

### 2.3. Preprocessing

Many sources of unwanted random radio frequency, electrical signals, and fluctuating voltages always exist around the proposed sources. The collected spectral data are then obtained by subtracting the background data with no fruit inside the box from the sample data before putting it into the machine learning system.

Figure 8 shows how the sweep background noise varied for different measuring positions. Figure 9 shows an example of the original spectra and the background noise cancellation spectra of a sample fruit.
(3)data=SweepObject−BackgroundNoise

### 2.4. Machine Learning Model

In this research, two kinds of machine learning were implemented. K-nearest neighbors (KNN) is one of the simplest machine learning algorithms, and is known as lazy learning. Each time we want to make a prediction, KNN searches for the nearest neighbor(s) in the entire training set. It does not have a training phase, and requires tuning only one hyper-parameter (the value of k). On the other hand, a neural network is more complex. It requires a training phase and involves many hyper-parameters controlling the size and structure of the network and the optimization procedure.

#### 2.4.1. KNN (K-Nearest Neighbor)

The first machine learning method is KNN (K-Nearest Neighbor), which is suitable when using a small dataset and is often the first choice of machine learning method. The KNN is a non-parametric method used for classification and regression. The K-Nearest Neighbor classifier usually applies either the Euclidean distance or the cosine similarity between the training tuples and the test tuple, but for the purpose of this research work, the Euclidean distance approach will be applied when implementing the KNN model for our recommendation system.

In our work, we applied the distance weighted KNN approach, in which we experimented with different values of K on our sample data, starting from K = 1, up to K = 9. We discovered that the experiment works best with K = 5, so we selected K = 5, which gives us the minimum error rate.

#### 2.4.2. Neural Network

In the neural network model, we use a signal neural perception for a simple model with configuration as below. The hyper-parameters of the neural network were selected after several rounds of testing.

Input Layer: series of 40 data points were sweep from 700 Mhz to 1 GHz;Hidden Layers: 1 hidden layer with 20 neural node;Activation function: tansig;Output Layer: 5 outputs in Dataset 1 and 10 outputs in Dataset 2.

The database processing and neural network model execution is done in the Matlab environment. Simulation is performed on a computer with the following configuration:Processor: Xeon E5-2630—2.40 GHz;RAM memory: 32 GB;GPU: NVIDIA TITAN V 12GB VGA;Environment: MATLAB R2018a.

#### 2.4.3. K-Fold Cross-Validation

Cross-validation is a re-sampling procedure used to evaluate machine learning models on a limited data sample. The value of K specifies the number of folds you plan to split the dataset into. Smaller values of K means that the dataset is split into fewer parts, but each part contains a larger percentage of the dataset. As such, the procedure is often called k-fold cross-validation. In our work, we choose a value of k = 10, which is very common in the field of applied machine learning.

## 3. Experimental Results

We carried out many experiments to evaluate the proposed classification algorithm and compare its performance with each other for fruit classifications. K-fold cross-validation was used to evaluate a model’s performance with two datasets. S-parameters, which include *S*_11_ and *S*_21_ with both magnitude and phase, are the input of the machine learning algorithms. The masured datasets are openly available on Ref. [16].

### 3.1. Experiments on Dataset 1

**Dataset 1:** In Dataset 1, 80 samples of 5 types of fruits were collected for classification, as described in Table 1. Then we evaluated the classification accuracy using KNN and neural network. We achieved the best accuracy of 98.75% and 99.75% with the KNN and neural network algorithm, respectively, on the test dataset using both *S*_11_ and *S*_21_ in Image parameter, as illustrated in Table 2. The confusion matrix in Table 3 shows the performance of the model on the test dataset. In the confusion matrices, the row represents the actual fruit class and the column represents the predicted fruit obtained by the proposed models.

### 3.2. Experiments on Dataset 2

**Dataset 2:** For extended investigation of fruit ripening, we developed another database, called Dataset 2 in Table 4, which contains 500 samples of 5 different types of fruits, including Avocado, Tomato, Orange, Guava, and Mango. Table 5 and Table 6 report the confusion matrix and classification accuracy on this dataset using KNN and neural network for comparison. It can be observed that the KNN model achieves a slightly higher correct classification rate on the test set compared to the neural network one (98.4% and 96.6%, respectively) for both *S*_11_ and *S*_21_ in Real parameter.

### 3.3. Comparison with Other Classification Methods

The following Table 7 presents the comparison between our proposed approach with the other approaches in terms of their accuracy, costs, speed, cost, and their disadvantages. It can be observed that the VNA approach achieves a slightly better recognition performance as compared to the NIR one for the same dataset. In terms of the recognition time, it took both of the two methods (NIR and VNA) from 2 to 3 ms to give the prediction on a fruit sample as well as ripeness recognition. Additionally, cost-effectiveness is another advantage of the VNA machine, whose price is only 60$ for the same identification purpose. These results confirm the potential of applying the VNA technique and machine learning technologies for fruit classification and ripeness classification.

## 4. Conclusions and Discussion

This paper presents a new technology adoption low-cost Vector Network analyzer and machine learning algorithms for sorting and grading fruits. For two existing KNN and NN models, we proposed feature extraction methods using four components of reflection coefficient *S*_11_ and transmission coefficient *S*_21_. The proposed models were evaluated on two separate datasets using five types of fruits, including Apple, Avocado, Dragon Fruit, Guava, and Mango. The experimental results showed that the neural network model achieved a slightly higher accuracy than the KNN one for fruit classification. On the other hand, KNN performed better than the other for the ripening dataset. For both applications, the two models achieved excellent accuracy on the two datasets, which shows the robustness of our novel approach. As for future work, the evaluation of the proposed framework will be extended by using extra fruit and vegetable species. A Faraday cage could be used to reduce the impact of environmental noise.

## Figures and Tables

**Figure 1 sensors-23-00952-f001:**
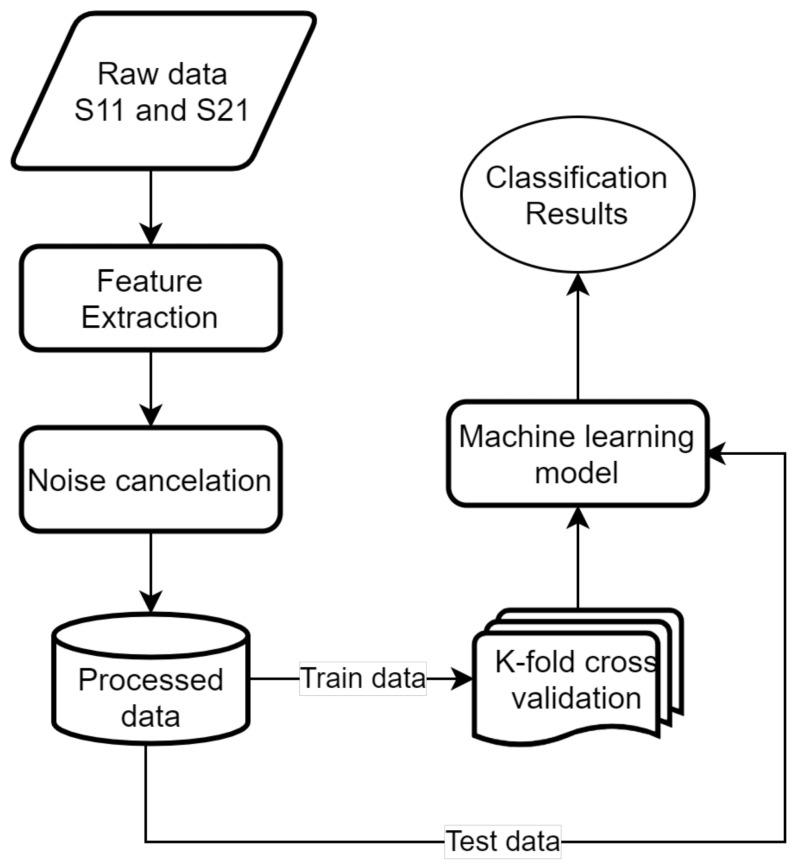
Technical steps diagram.

**Figure 2 sensors-23-00952-f002:**
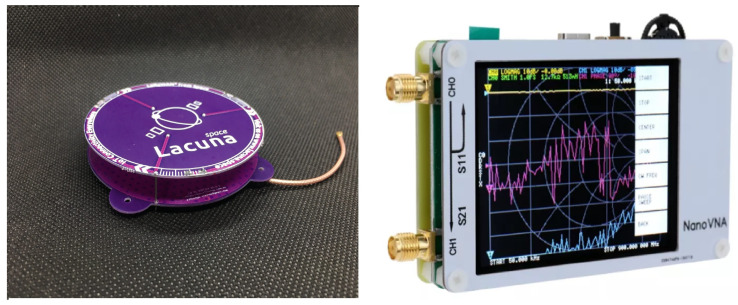
Measurement components.

**Figure 3 sensors-23-00952-f003:**
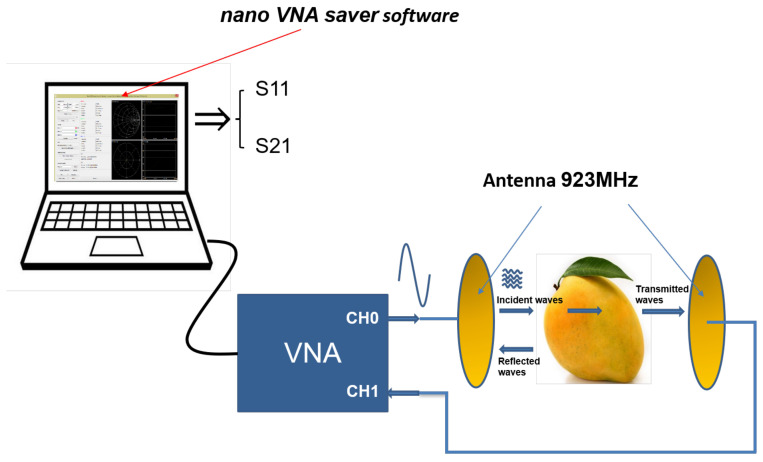
Data collection model.

**Figure 4 sensors-23-00952-f004:**
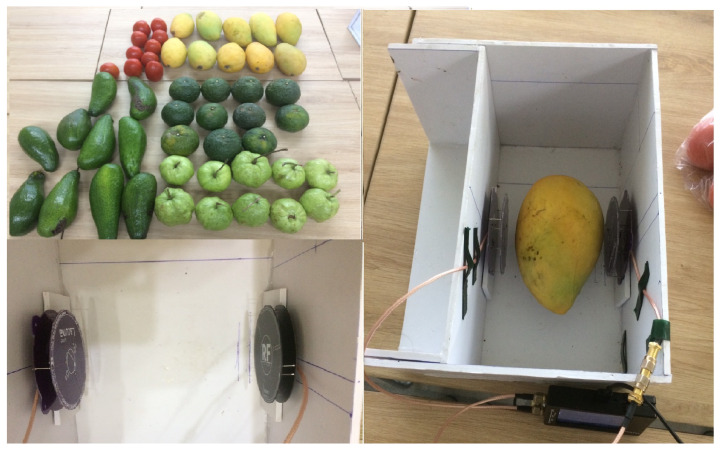
Data collection.

**Figure 5 sensors-23-00952-f005:**
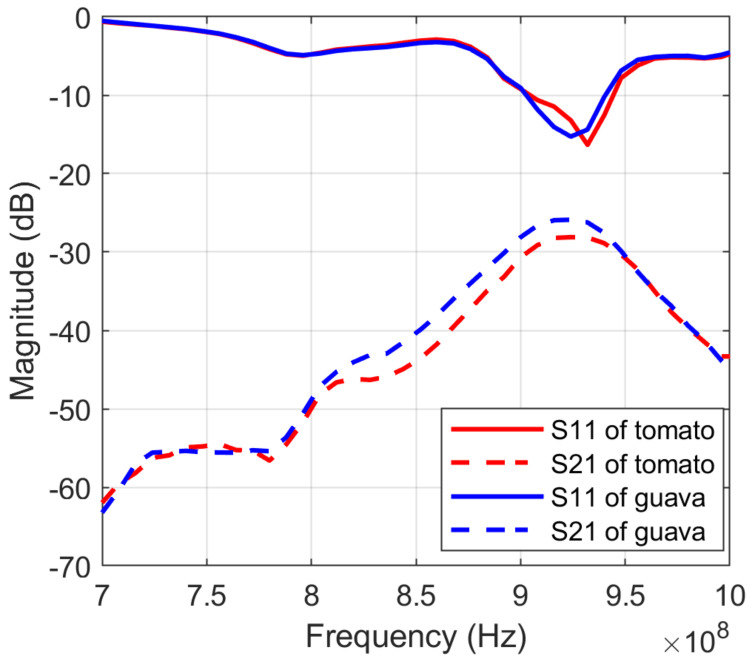
*S*_11_ and *S*_21_ of tomato and guava.

**Figure 6 sensors-23-00952-f006:**
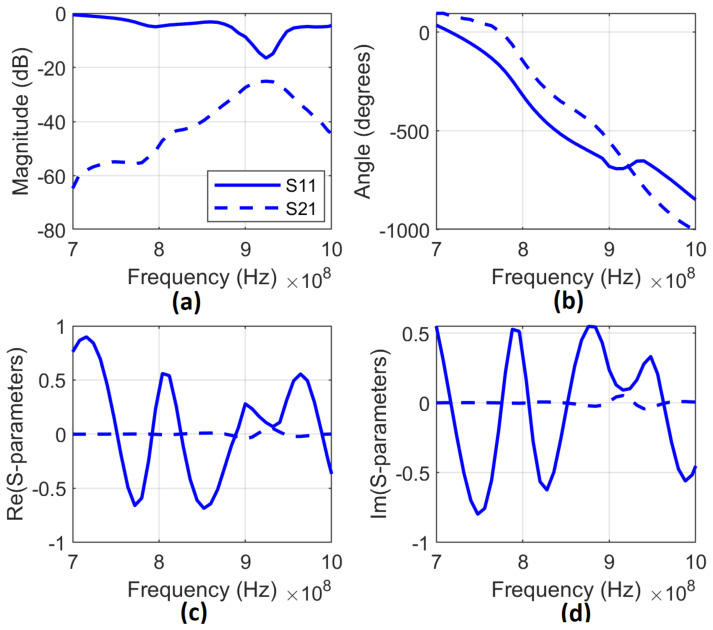
Magnitude (**a**), Phase (**b**), Real (**c**), and Image (**d**) of *S*_11_ and *S*_21_.

**Figure 7 sensors-23-00952-f007:**
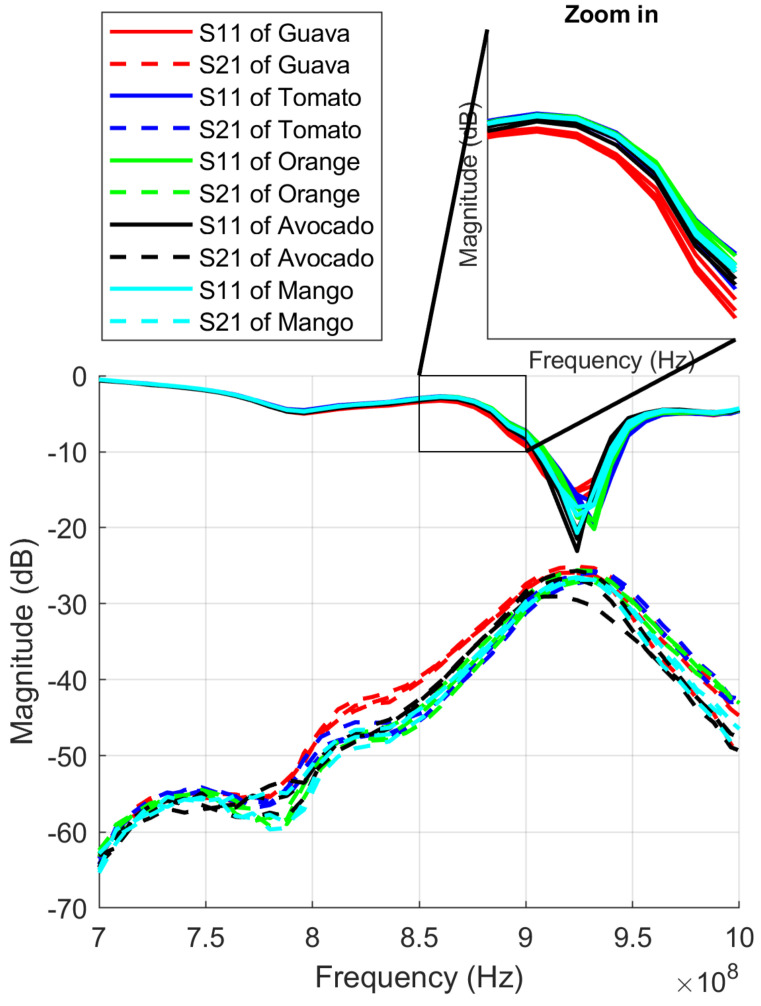
*S*_11_ and *S*_21_ of fruits.

**Figure 8 sensors-23-00952-f008:**
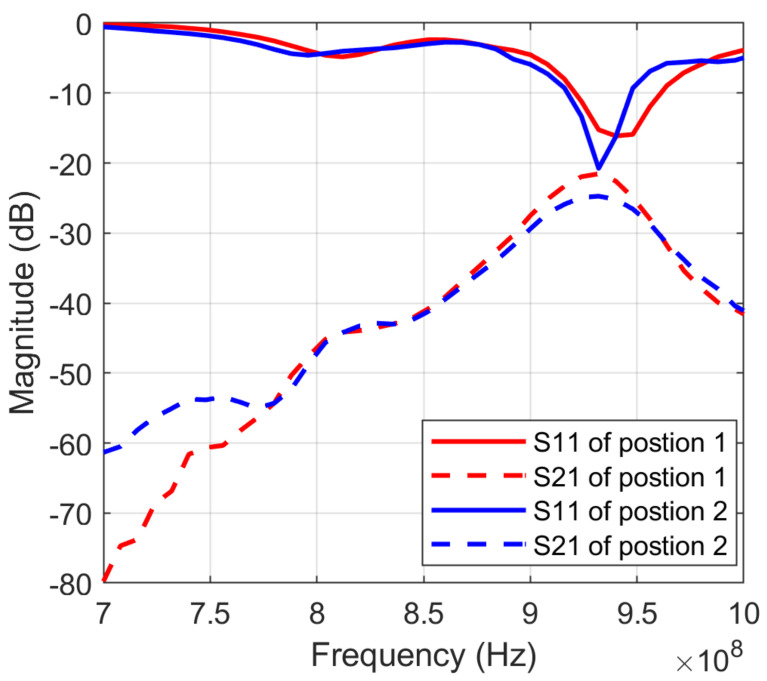
*S*_11_ and *S*_21_ of background noise in two different locations with positions 1 and 2.

**Figure 9 sensors-23-00952-f009:**
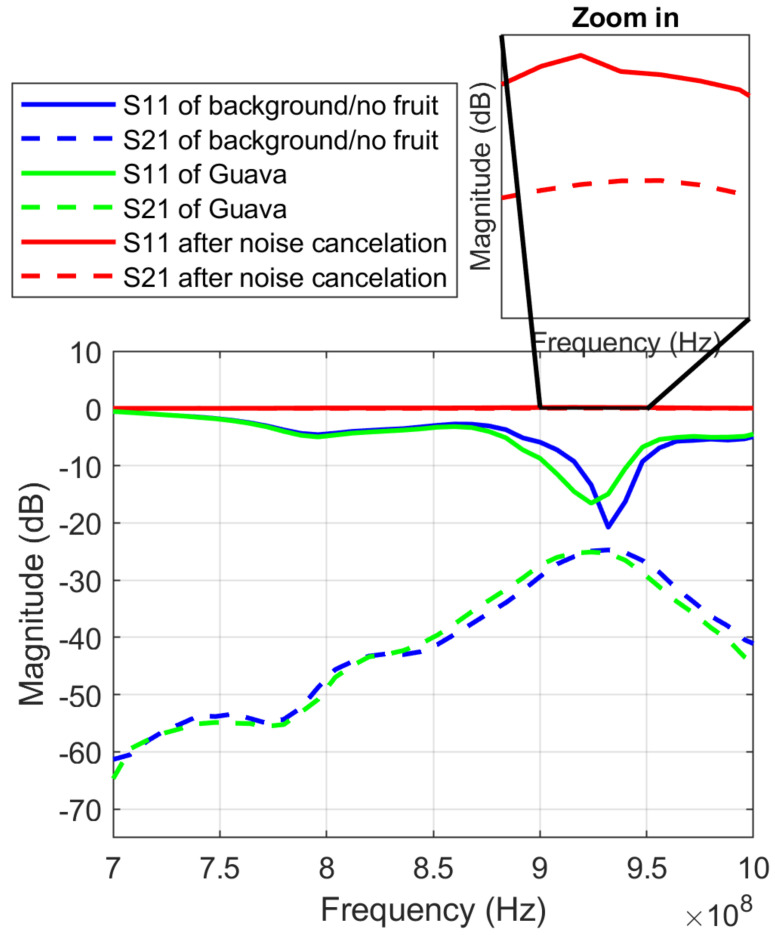
Data after noise cancellation.

**Table 1 sensors-23-00952-t001:** Dataset 1.

Fruit	Quantity	Size (Diameter)	Sweep
Orange	80	7 cm +/−1 cm	700–1500
Guava	80	7 cm +/−2 cm	700–1500
Tomato	80	5 cm +/−1 cm	700–1500
Avocado	80	15 cm +/−3 cm	700–1500
Mango	80	9 cm +/−2 cm	700–1500

**Table 2 sensors-23-00952-t002:** Classification accuracy.

KNN			
Feature	*S* _11_	*S* _21_	*S*_11_ + *S*_21_
Amplitude	95%	95.25%	96%
Real	95.75%	98.5%	98.25%
Image	98%	98.25%	**98.75**%
Phase	89.25%	91.75%	94.75%
**Neural Network**			
Feature	*S* _11_	*S* _21_	*S*_11_ + *S*_21_
Amplitude	97.25%	95.75%	99.25%
Real	97.75%	99%	99%
Image	98.25%	99.5%	**99.75**%
Phase	91%	90.75%	96%

**Table 3 sensors-23-00952-t003:** Dataset 1—confusion matrix.

**KNN**					
	Avocado	Tomato	Orange	Guava	Mango
Avocado	75	0	0	3	2
Tomato	0	80	0	0	0
Orange	0	0	80	0	0
Guava	0	0	0	80	0
Mango	0	0	0	0	80
**Neural Network**					
	Avocado	Tomato	Orange	Guava	Mango
Avocado	80	0	0	0	0
Tomato	0	79	0	1	0
Orange	0	0	80	0	0
Guava	0	0	0	80	0
Mango	0	0	0	0	80

**Table 4 sensors-23-00952-t004:** Dataset 2.

Fruit	Quantity	Antenna	Sweep
		Mhz	Mhz
Ripe Avocado	50	923	700–1500
Avocado	50	923	700–1500
Ripe Tomato	50	923	700–1500
Tomato	50	923	700–1500
Yellow Orange	50	923	700–1500
Green Orange	50	923	700–1500
Ripe Guava	50	923	700–1500
Guava	50	923	700–1500
Mango	50	923	700–1500
Ripe Mango	50	923	700–1500

**Table 5 sensors-23-00952-t005:** Dataset 2—confusion matrix.

**KNN**										
	Ripe Avocado	Avocado	Ripe Tomato	Tomato	Yellow Orange	Green Orange	Ripe Guava	Guava	Mango	Ripe Mango
Ripe Avocado	47	2	0	0	0	0	0	1	0	0
Avocado	1	48	0	0	0	0	0	0	0	1
Ripe Tomato	0	0	50	0	0	0	0	0	0	0
Tomato	0	0	0	50	0	0	0	0	0	0
Yellow Orange	0	0	0	0	50	0	0	0	0	0
Green Orange	0	0	0	0	0	50	0	0	0	0
Ripe Guava	0	0	0	0	0	0	50	0	0	0
Guava	0	0	0	0	0	0	0	50	0	0
Mango	0	1	0	0	0	0	0	0	48	1
Ripe Mango	0	0	0	0	0	0	0	0	1	49
**Neural Network**										
	Ripe Avocado	Avocado	Ripe Tomato	Tomato	Yellow Orange	Green Orange	Ripe Guava	Guava	Mango	Ripe Mango
Ripe Avocado	46	2	1	0	0	0	1	0	0	0
Avocado	0	47	0	0	0	0	1	0	0	2
Ripe Tomato	0	1	49	0	0	0	0	0	0	0
Tomato	0	0	0	48	0	0	0	1	0	1
Yellow Orange	0	0	0	0	50	0	0	0	0	0
Green Orange	0	0	0	0	0	50	0	0	0	0
Ripe Guava	1	0	0	0	0	0	48	1	0	0
Guava	0	0	0	0	0	0	0	50	0	0
Mango	1	1	0	0	0	0	0	0	46	3
Ripe Mango	0	0	0	0	0	0	0	1	0	49

**Table 6 sensors-23-00952-t006:** Classification accuracy.

**KNN**			
Feature	*S* _11_	*S* _21_	*S*_11_ + *S*_21_
Amplitude	80.2%	93.2%	93.6%
Real	88%	97.2%	**98.4**%
Image	89.2%	96.4%	96.6%
Phase	66.8%	83.4%	85.6%
**Neural Network**			
Feature	*S* _11_	*S* _21_	*S*_11_ + *S*_21_
Amplitude	77.2%	83.2%	92.2%
Real	88.2%	95.2%	**96.6**%
Image	90.2%	95%	95.4%
Phase	67.6%	81.6%	80%

**Table 7 sensors-23-00952-t007:** Comparison with other methods.

Method	Accuracy	Speed	Cost	Advantage	Drawback
VNA	99.75%	fast	∼60$	sense inside of the fruit	sensitive to RF
NIR [17]	99%	fast	∼1200$	sense surface of the fruit	sensitive to sunlight
Image processing [8]	99.75%	slow	∼100$	sense color and shape of fruit	sensitive to lighting conditions

## Data Availability

Dataset from Measurements. Available online: https://github.com/Lic-Tran-Van/FruitClassification (accessed on 8 November 2022).

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
