# Peer review of "The Novel Combination of Nano Vector Network Analyzer and Machine Learning for Fruit Identification and Ripeness Grading"

_sensors, 2023, doi:10.3390/s23020952_

Round 1

Author Response

We would like to thank the reviewers and editors for their valuable comments that help to improve the manuscript's quality.
You will find in the attached document a detailed answer to all reviewer comments and a new version of the manuscript with changes highlighted in blue.

Reviewer 2 Report

In this work, the authors proposed a machine learning approach to classify fruits. The results are promising. However, there are several concerns to address before publication of the paper:

1. Why k-NN and Neural Net? There lack enough motivations or experiments to choose these two models. For example, why not decision tree/random forest or support vector machine? The authors should provide more explanation or numerical comparison to justify their choice.

2. Hyper-parameter tuning. The authors only provide a set of parameters of neural network for their modeling. Is it the only one? Why not try different hyper-parameters?

3. The authors should make their dataset and trained models openly available for other researchers to repeat their works.

Author Response

(The authors gave the same response as above.)

Round 2

Reviewer 1 Report

Comments are mostly well addressed. No further comments.

Reviewer 2 Report

The authors have addressed my concerns. I recommend publication.